# Comprehensive genome-wide association study of different forms of hernia identifies more than 80 associated loci

João Fadista[1,2,3], Line Skotte [1], Juha Karjalainen[3,4,5], Erik Abner [6], Erik Sørensen[7], Henrik Ullum[7,8], Thomas Werge [9,10,11,12], iPSYCH Group*, Tõnu Esko[6], Lili Milani [6], Aarno Palotie [3,4,13], Mark Daly[3,4], FinnGen Consortium, Mads Melbye [10,14,15,16,23], Bjarke Feenstra [1,23] & Frank Geller [1,23 ✉]

Hernias are characterized by protrusion of an organ or tissue through its surrounding cavity and often require surgical repair. In this study we identify 65,492 cases for five hernia types in the UK Biobank and perform genome-wide association study scans for these five types and two combined groups. Our results show associated variants in all scans. Inguinal hernia has the most associations and we conduct a follow-up study with 23,803 additional cases from four study groups giving 84 independently associated variants. Identified variants from all scans are collapsed into 81 independent loci. Further testing shows that 26 loci are associated with more than one hernia type, suggesting substantial overlap between the underlying genetic mechanisms. Pathway analyses identify several genes with a strong link to collagen and/or elastin (*ADAMTS6*, *ADAMTS16*, *ADAMTSL3*, *LOX*, *ELN*) in the vicinity of associated loci for inguinal hernia, which substantiates an essential role of connective tissue morphology.

[1] Department of Epidemiology Research, Statens Serum Institut, Copenhagen, Denmark. [2] Department of Clinical Sciences, Lund University Diabetes Centre, Malmö, Sweden. [3] Institute for Molecular Medicine Finland, FIMM, HiLIFE, University of Helsinki, Helsinki, Finland. [4] Broad Institute of MIT and Harvard, Cambridge, MA, USA. [5] Analytic and Translational Genetics Unit, Massachusetts General Hospital and Harvard Medical School, Boston, MA, USA. [6] Estonian Genome Center, Institute of Genomics, University of Tartu, Tartu 51010, Estonia. [7] Department of Clinical Immunology, Copenhagen University Hospital, Rigshospitalet, Copenhagen, Denmark. [8] Statens Serum Institut, Copenhagen, Denmark. [9] iPSYCH, The Lundbeck Foundation Initiative for Integrative Psychiatric Research, Aarhus, Denmark. [10] Department of Clinical Medicine, University of Copenhagen, Copenhagen, Denmark. [11] Lundbeck Foundation Center for GeoGenetics, GLOBE Institute, University of Copenhagen, Copenhagen, Denmark. [12] The Lundbeck Foundation Initiative for Integrative Psychiatric Research (iPSYCH), Copenhagen and Aarhus, Aarhus, Denmark. [13] Psychiatric & Neurodevelopmental Genetics Unit, Department of Psychiatry, Analytic and Translational Genetics Unit, Department of Medicine, and the Department of Neurology, Massachusetts General Hospital, Boston, MA, USA. [14] K.G. Jebsen Center for Genetic Epidemiology, Norwegian University of Science and Technology, Trondheim, Norway. [15] Norwegian Institute of Public Health, Oslo, Norway. [16] Department of Genetics, Stanford University School of Medicine, Stanford, CA, USA. [23] These authors contributed equally: Mads Melbye, Bjarke Feenstra, Frank Geller. *A list of authors and their affiliations appears at the end of the paper. A list of members and their affiliations appears in the Supplementary Information. ✉email: fge@ssi.dk

A hernia is an outpouching of tissue through a preformed or secondarily established fissure[1]. Most hernias occur in the abdominal wall, only diaphragmatic hernias present with a defect of the diaphragm resulting in abdominal tissue entering the chest cavity. A Swedish study on the familial risk of different hernias, including inguinal, umbilical, femoral and incisional hernias, identified increased sibling risks not only for pairs with the same form of hernia, also for several combinations of discordant hernias[2]. They concluded that genetic factors play a role in the etiology of hernias. The fact that increased risks were seen across different forms of hernia supports the hypothesis that there might be genetic risk factors contributing to multiple forms of hernia and combined analyses have potential to identify additional associated loci. Higher BMI is associated with increased and decreased risk of diaphragmatic hernia[3] and inguinal hernia[4], respectively. On the other hand, inguinal hernia and gastroesophagal reflux (a frequent consequence of diaphragmatic hernia) have been associated with psychiatric disorders later in life, including depression and schizophrenia[5,6]. Comprehensive GWAS data on hernias are needed to investigate potential shared genetic susceptibility between hernias and these and other traits. So far, the only genome-wide association study on hernias was on inguinal hernias and identified four risk loci[7].

In this genome-wide association study based on data from the UK Biobank[8], we investigate five different forms of hernia classified according to ICD 10 codes by their location as inguinal (K40), femoral (K41), umbilical (K42), and diaphragmatic (K44), or by their origin in previous injury of the abdominal wall (ventral hernia, K43). To maximize power, we combine information about hernia from hospital records (surgery codes, ICD9- and ICD10-codes) and interview data (surgery and non-cancer illness). We also study two combined groups: any hernia and any hernia excluding individuals who only had diaphragmatic hernia, as diaphragmatic hernia is special by its location and the substantially lower rate of surgery-confirmed cases. Variants associated with inguinal hernia are followed up in silico in four additional study groups. In total, we identify 81 genetic loci associated with one or multiple forms of hernia. Pathway analysis provides further evidence that genes important for connective tissue morphology play an essential role in inguinal hernia etiology.

## Results

**Genome-wide association scans in the UK Biobank.** Figure 1 presents the design of the study, including sample sizes for the different forms of hernia in the UK Biobank and the set-up of the inguinal hernia follow-up study. We performed genome-wide association studies (GWAS) scans of all individuals with British ancestry confirmed by genetic principal components analysis for the seven hernia groups with SAIGE[9], a method that adjusts for case-control imbalance and relatedness amongst individuals. Some individuals had more than one form of hernia and the correlation between the different hernia groups (Supplementary Data 1) ranged from $r = 0.006$ (95%-CI: 0.003–0.009) between diaphragmatic and femoral to $r = 0.133$ (95%-CI: 0.130–0.136) between umbilical and ventral. The combined groups were strongly correlated (all $r > 0.6$) with the large hernia groups they included (Supplementary Data 1). We performed the analyses for the full group of cases and controls (*all* analysis), as well as for the subgroups of only male or female cases and controls (male/female analysis). For all GWAS scans we used individuals without any form of hernia as controls. $P$ values were adjusted by genomic control, ranging from 1.00 (femoral) to 1.18 (any hernia), and the increase in genomic control was largely driven by common variants (minor allele frequency ≥ 5%, Supplementary Data 1), which also accounted for the vast majority of variants with $P < 5 \times 10^{-8}$.

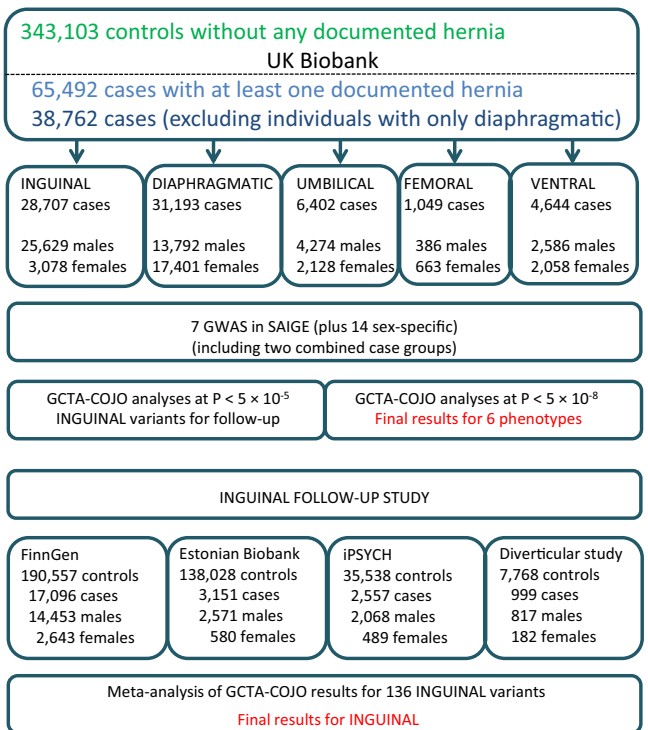

**Fig. 1 Outline of this study.** Study design and sample sizes for the study groups.

All seven scans for the different hernia groups identified genetic variants at genome-wide significance (defined per analyzed phenotype as $P < 5 \times 10^{-8}$). Figure 2 shows Manhattan plots for the seven analyzed phenotypes (see Supplementary Fig. 1 for the respective quantile–quantile plots). We did a conditional and joint analysis of the initial SAIGE results with GCTA-COJO[10] to also investigate multiple hits per locus, and we observed between two and 55 independently associated variants per phenotype at $P < 5 \times 10^{-8}$ (Supplementary Data 2). There was overlap between the identified loci per hernia type and the combined groups identified additional loci. A genetic link between the different forms of hernia was further supported by genetic correlation analysis of the five scans for specific forms of hernia, where we observed significantly increased $r_g$'s for all combinations ranging from 0.24 (95%-CI: 0.17–0.31) between inguinal and diaphragmatic to 0.76 (95%-CI: 0.56–0.95) between femoral and ventral (Supplementary Data 1).

The sex-specific analyses mainly identified associated variants at $P < 5 \times 10^{-8}$ in the male analyses and most overlapped with loci identified in the *all* analyses for these phenotypes (Supplementary Data 2). Furthermore, the analyses showed that the most significantly associated loci can differ between males and females (see Supplementary Fig. 2 for Miami plots of the sex-specific scans).

**Follow-up study of variants associated with inguinal hernia.** For inguinal hernia, we followed up associated variants in silico in the FinnGen project, the Estonian Biobank and two Danish cohorts, providing results for almost 24,000 cases and more than 370,000 controls (Fig. 1). The GCTA-COJO analyses identified several regions with multiple associated variants. Therefore, we used the results from GCTA-COJO analyses of inguinal hernia at $P < 1 \times 10^{-5}$ as starting point and tried to identify these variants or surrogate SNPs with an $r^2 > 0.8$ in the four follow-up groups. Our goal was to achieve the genome-wide significance threshold $5 \times 10^{-8}$ in the combined analysis of the UK biobank and the four follow-up studies. Of the 136 investigated variants, only three had

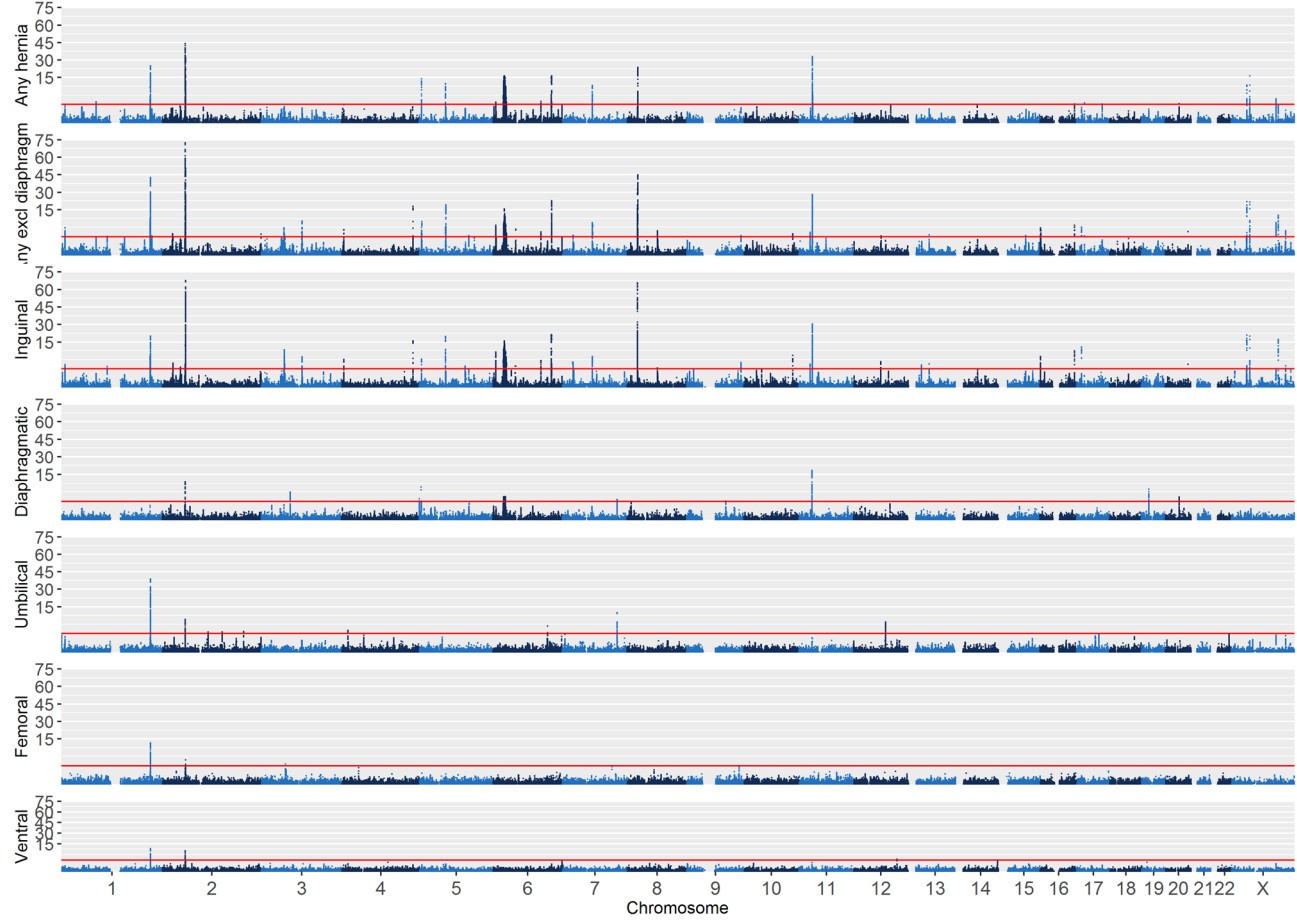

**Fig. 2 Manhattan plots for SAIGE GWAS scans.** Manhattan plots for the 7 analyzed hernia groups, displaying $-\log_{10}$ ($P$ values) from the SAIGE analyses over the chromosomes. The scale for the y-axis is shrinked by a factor of 5 for values above 15. The red line indicates the $P$ threshold for genome-wide significance of $5 \times 10^{-8}$.

no information in all four follow-up groups, while two more were missing in both FinnGen and Estonian Biobank. After performing GCTA-COJO analyses in the follow-up groups, the results were meta-analyzed with METAL[11]. The number of variants reaching genome-wide significance increased to 84 (from 53 in UK Biobank alone) in all cases (Supplementary Data 3). In the combined analysis of the four follow-up studies alone, 50 out of these 84 variants had a $P$ below the strict Bonferroni threshold adjusting for 133 variants ($3.76 \times 10^{-4}$), 31 had a $P < 0.05$, another variant had a $P$ of 0.08 and the last two variants with $P > 0.1$ were from the HLA-locus, a region well known for a complicated LD structure that nevertheless had one variant at $P < 5 \times 10^{-8}$. Applying a strict replication approach for the 41 variants with $P < 7.14 \times 10^{-9}$ (adjusting for 7 analyzed phenotypes without taking the correlation between them into account), would yield 33 variants reaching the significance threshold of $1.22 \times 10^{-3}$ (0.05/41) in the combined follow-up studies. Applying a conservative replication approach demanding a $P < 7.14 \times 10^{-9}$ (adjusting for 7 analyzed phenotypes without taking the correlation between them into account, reached by 41 variants) in the UK Biobank GWAS scan, and a $P < 1.22 \times 10^{-3}$ (0.05 divided by the 41 variants taken forward) in the combined follow-up studies, would result in 33 variants fulfilling both of these criteria.

In the sex-specific analyses, the number of variants at $P < 5 \times 10^{-8}$ increased to 70 (of which 47 had reached the threshold in the UK Biobank) in males and to 7 (4) in females (Supplementary Data 3). As expected, there was a large overlap between the *all*

and the sex-specific findings, and some of the hits were already identified in the UK Biobank larger groups combining any hernia or any hernia excluding diaphragmatic. The smallest follow-up study group was based on a Danish study of diverticular disease, a condition with associated variants also showing association with different forms of hernia, including inguinal[12]. A sensitivity analysis with regard to the diverticular disease study did not show substantial differences in the results (Supplementary Note 1).

**Summary of all identified loci.** The loci identified for the different hernia groups partially overlapped, so we decided to collapse loci by binning variants with an $r^2 > 0.6$ under a key SNP with the lowest $P$ value in any of the UK Biobank GCTA-COJO analyses. For loci we chose a broad definition of 500 kb to the next associated key SNP and for key SNPs with $P < 1 \times 10^{-30}$ the distance was increased to 1 Mb as changes in the conditional $P$ values were observed even over these distances; the HLA region on chromosome 6 was considered one locus. We added new variants identified in the inguinal meta-analyses after the initial binning.

In total, the analyses of the *all* groups resulted in 81 collapsed loci with 114 distinct key SNPs, the sex-specific analyses added another 5 loci (4 in males) and 21 key SNPs (Supplementary Data 3). The findings were still concentrated in the groups including inguinal hernia and for 69 of the 81 loci at least one key SNP showed the lowest $P$ value for inguinal hernia or one of the combined groups including inguinal hernia, the inguinal hernia

follow-up study alone accounted for 22 of these loci. Up to 6 distinct key SNPs were identified per locus; sometimes multiple key SNPs for the same phenotype, sometimes there was variation regarding the associated phenotypes or sexes. The larger number of sex-specific findings in males was at least partly driven by the higher statistical power due to larger numbers of male cases, especially for inguinal hernia. Five key SNPs were identified in the HLA region and subsequent comparison with HLA-alleles showed that none of these SNPs was a best match for a known HLA-allele, only one HLA allele showed an $r^2 > 0.5$ for a key SNP (HLA-B*8:01 with rs9281226, $r^2 = 0.67$).

We investigated heterogeneity between the sexes for the associated variants identified at genome-wide significance by meta-analyzing the results for males and females with METAL[11]. The analysis was restricted to variants genome-wide significant for specific forms of hernias as the composition of the combined groups largely differed between the sexes (Supplementary Data 1). Among 110 variants tested, 50 variants showed no indication for heterogeneity ($I^2 = 0$) so that analyzing both sexes together had increased statistical power substantially (Supplementary Data 3). For 22 (21 inguinal) variants a high degree of heterogeneity ($I^2 > 75$) was observed. However, this did not necessarily mean that the association was confined to one sex: 10 variants showed no indication of association with inguinal hernia in females (effect in other direction or $P > 0.2$), but in the other 12 instances the analysis for the other sex reached at least nominal significance (range of higher sex-specific $P$: $1.6 \times 10^{-3}$ to $1.9 \times 10^{-14}$, for 10 of these variants the stronger effect estimates were observed in females).

**MultiPhen and variants with effects in opposite directions in different types of hernia.** As some variants were associated with more than one hernia group and the two combined hernia groups only cover a small fraction of all combinations of hernia types, we decided to pinpoint the associated hernia types by running MultiPhen[13] on the 114 associated key SNPs (Supplementary Data 3). For a specific variant, a hernia type was considered associated if it was associated at $P < 0.01$ and the results were summarized by locus, i.e. a hernia group was considered associated if any of the variants was associated. The summary of the associated hernia combinations for the 81 loci can be seen in Fig. 3. A locus on chromosome 7q33 (key SNP rs12707188) displayed risk for umbilical and diaphragmatic hernia at genome-wide significance for two variants in LD ($r^2 = 0.76$). Checking the effect in the other phenotype showed that the risk allele for umbilical hernia was protective for diaphragmatic and vice versa (Supplementary Data 3). The variants are located in *CALD1* (calmodulin- and actin-binding protein), a gene important for smooth muscle and nonmuscle contraction. Studies on the molecular level are needed to investigate whether functional changes in this gene indeed mean opposite risk outcomes for the two forms of hernia.

This finding lead us to investigate the potential of opposite effects on a broader scale by analyzing all variants with $P < 0.001$ for at least one of the seven phenotypes under study with MultiPhen. We only considered variants outside the previously investigated collapsed loci and confirmed the initial results by linear regression in R[14] on a subset of 335,258 unrelated individuals. Here we identified five new suggestive loci, four of which displayed effects going opposite directions, three times the hernia combination was inguinal and diaphragmatic and one time inguinal, umbilical and ventral (going the same direction as inguinal); the remaining locus had a main effect for umbilical seen at a lower level for diaphragmatic hernia. The five suggestive loci are displayed in Supplementary Data 3.

To pinpoint genes and pathways responsible for the observed association findings and to investigate functional implications on a broader scale, we ran several pathway analysis tools on our results.

**MAGMA/FUMA.** We applied gene prioritization by MAGMA[15] as implemented in FUMA[16]; genes were considered associated if $P < 2.55 \times 10^{-6}$, the Bonferroni correction considering 19,622 genes analyzed. Genes were prioritized for all seven analyzed phenotypes, with the largest numbers observed for inguinal hernia and the two combined groups (Supplementary Data 4).

FUMA expression results for 54 GTEx tissues were derived from MAGMA analysis. For inguinal hernia 17 tissues reached the group-wise significance ($P < 9.26 \times 10^{-4}$) including both adipose tissues, cultivated fibroblast cells, and the esophagus muscle tissue, the findings in the two combined groups were very similar (Supplementary Data 4). Additional findings were seen for diaphragmatic hernia (artery aorta) and umbilical hernia (small intestine/terminal ileum and transverse colon).

**DEPICT.** We also performed DEPICT[17] analyses for each of the seven analyzed phenotypes based on the associated variants with $P < 1 \times 10^{-5}$ as input. Thus, the gene prioritization is targeted at genes in the vicinity of these variants. Genes with a false discovery rate (FDR) below 5% were only found for inguinal hernia and the two combined groups (Supplementary Data 5). For inguinal hernia 64 of the 196 investigated genes had an FDR < 5% and we will take a closer look at these genes in the Discussion.

Similarly, in the DEPICT analyses of 10,968 genesets, we detected 451 sets with an FDR < 5% for inguinal hernia, and somewhat lower numbers for the two combined groups, whereas the other four hernia types had no findings (Supplementary Data 5). The geneset descriptions had many keywords like abnormal, morphology, morphogenesis, development, muscle, embryo, tissue, and renal/kidney. Supplementary Fig. 3 illustrates the findings with 36 meta genesets derived by clustering the significantly enriched genesets, and gives the meta geneset represented by embryonic morphogenesis as an example.

Enrichment of expression in particular tissues and cell types was studied by testing whether genes in associated regions were highly expressed in any of 209 Medical Subject Heading (MeSH) annotations in data from 37,427 microarrays (Supplementary Data 5). In total, 36 significant tissue or cell type annotations among the 209 analyzed categories were identified for any hernia (34), umbilical, and any hernia excluding diaphragmatic (one each). The findings for any hernia substantiate trends observed in the analysis of inguinal hernia, the hits were mainly in the musculoskeletal, urogenital, and cardiovascular systems, connective and muscle tissue cells and connective and membrane tissues (Fig. 4).

**Gene Network.** Gene Network prioritizes genes based on expression data analysis specific for diseases based on human phenotype ontology (HPO) terms[18]. We compared results for the term "inguinal hernia" (HP: 0000023) in the 192 genes from the DEPICT gene prioritization for inguinal hernia with the referring DEPICT results (Supplementary Data 5). Nine of the genes showed significant results when adjusting for the 192 tests, 8 of which also had FDR < 5% in the DEPICT gene prioritization and 7 of the genes were in collapsed loci reaching genome-wide significance in our study.

**GARFIELD.** We applied GARFIELD (version 2)[19] to assess the relative enrichment of phenotype–genotype associations to a generic regulatory annotation denoting open chromatin in 424 cell lines and primary cell types from ENCODE and Roadmap Epigenomics. The hotspots analyses of the DNaseI hypersensitive

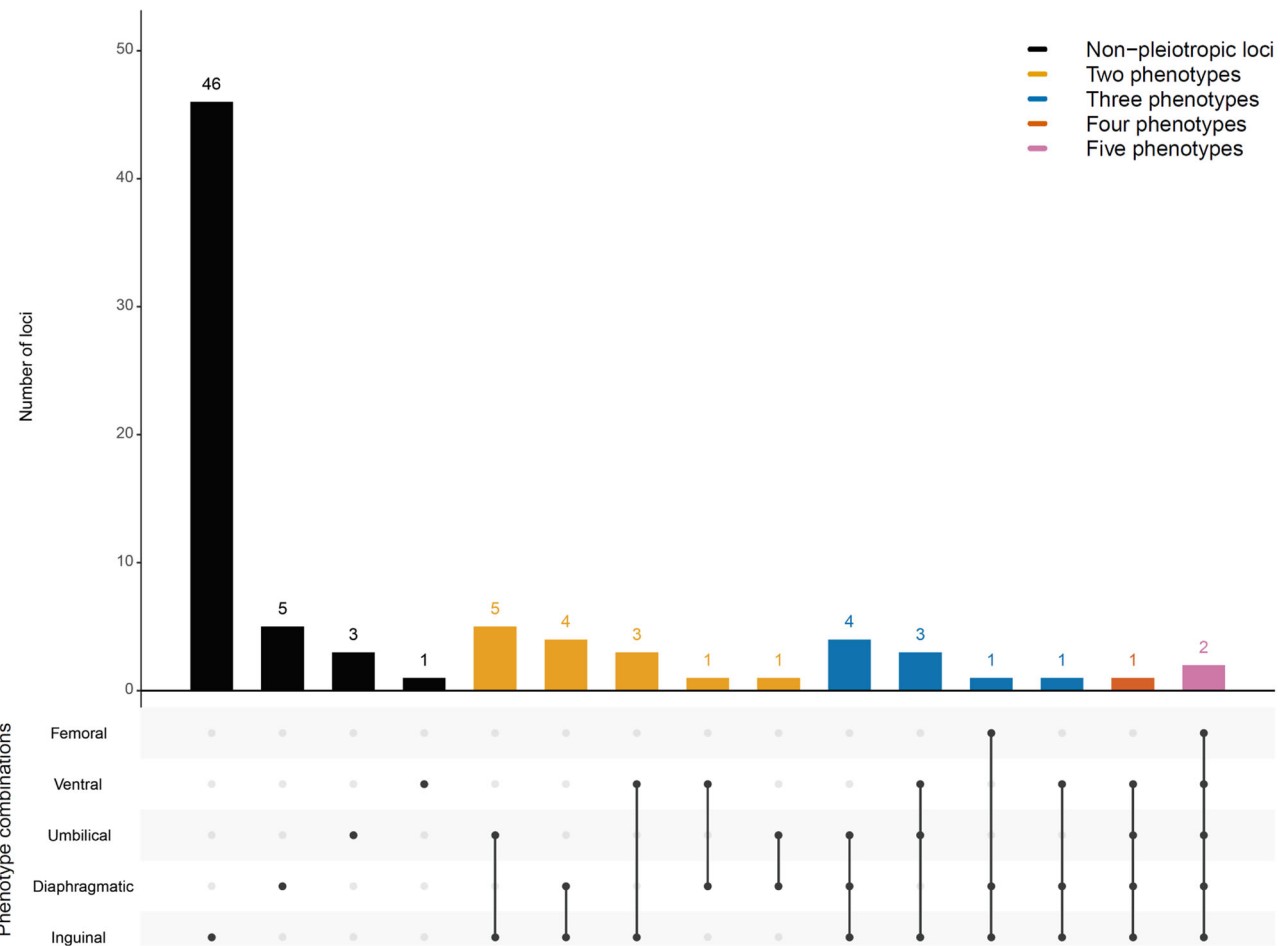

**Fig. 3 MultiPhen results for associated loci.** Distribution of the associated hernia traits (a trait is considered associated if $P < 0.01$ in the MultiPhen analysis, adjusting for 5 analyzed phenotypes) for the 81 associated loci.

sites identified by far the most associations (for complete results for the 7 analyzed phenotypes see Supplementary Data 6). Figure 5 illustrates the ubiquitous enrichment seen for inguinal hernia in an enrichment wheel plot. Many of the analyzed tissues were of fetal origin, and for several of them the enrichment was striking: at the level of $P < 1 \times 10^{-5}$, all 48 fetal muscle and all 11 fetal heart tissues had an empirical $P < 1.18 \times 10^{-4}$ (threshold adjusting for the 424 investigated cell types), another group with many enriched tissue probes was fibroblast, with all 19 tissues below the threshold.

**Gene expression in mice.** We checked genes identified for hernia phenotypes in mice (Mammalian Phenotype Ontology Annotations: http://www.informatics.jax.org/vocab/mp_ontology, data 10/31/2019), for overlap with genes annotated in the DEPICT analyses. In one mouse strain (C57BL/6), Efemp1(−/−) mice developed multiple large hernias including inguinal hernias, pelvic prolapse, and protrusions of the xiphoid process, additional histological analysis revealed a marked reduction of elastic fibers in fascia[20]. In our analyses, the locus containing *EFEMP1* showed association with all five forms of hernia. However, the function of the encoded protein in humans is unknown.

**LD score regression, Mendelian randomization, heritability.** Figure 6 shows the significant findings from bivariate LD score regression between the different forms of hernia and 245 traits (Supplementary Data 7), with the majority of findings for

diaphragmatic hernia. In connection with the iPSYCH project[21], we took a closer look at six psychiatric disorders and the finding for depressive symptoms and diaphragmatic hernia was also seen for major depressive disorder, both with diaphragmatic ($r_g = 0.23$, 95%-CI: 0.10–0.35) and ventral hernia ($r_g = 0.45$, 95%-CI: 0.19–0.71), all other correlations were not significant considering the 30 tests involved (Supplementary Data 7). Following up the findings for major depressive disorder, we performed Mendelian randomization using associated genetic variants from a recent GWAS of major depressive disorder[22] as instrumental variables and the five forms of hernia as outcome (Supplementary Note 2, Supplementary Data 7). The results showed a potential causal effect of the major depressive disorder instrumental variables on diaphragmatic hernia and to a lesser extent on ventral hernia. The analyses also indicated horizontal pleiotropy for some genetic variants, so that further studies are needed to clarify the effect.

We estimated heritability for the five forms of hernia on the liability scale with three different methods (Supplementary Data 7). The estimates from LD score regression[23] were 12.3% (95% CI: 9.8–14.7%) for inguinal hernia and ranging from 9.0% (95% CI: 5.8–12.3%) for ventral hernia to 12.8% (95% CI: 9.5–16.1%) for umbilical hernia. The estimates from the other two methods were lower in most instances.

**Discussion**
We performed GWAS scans on five different forms of hernia and two combined groups of any hernia and any hernia excluding diaphragmatic. Each individual scan resulted in at least two

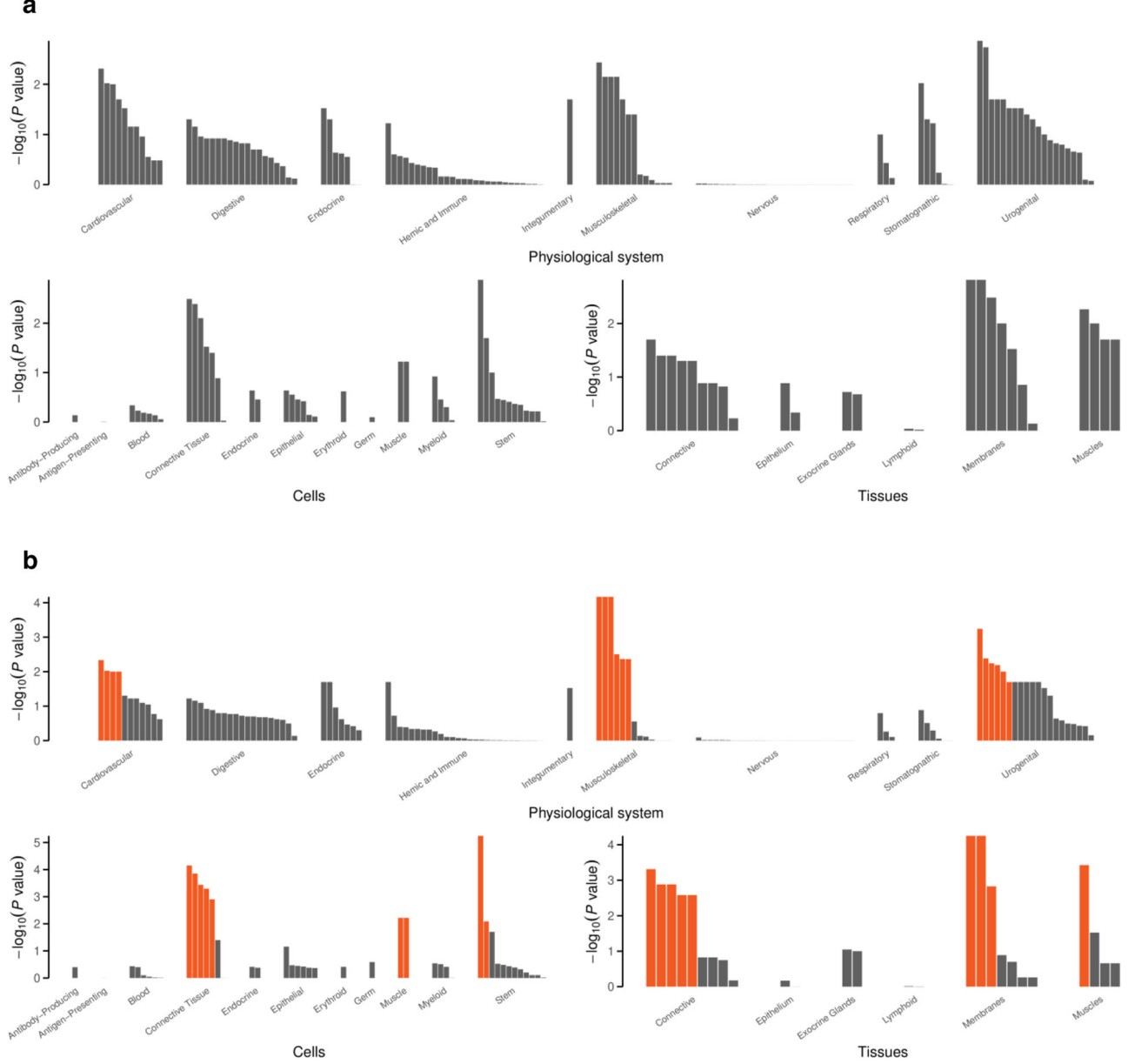

**Fig. 4 DEPICT enrichment plots.** Plots showing the enrichment of loci associated with **a** inguinal hernia and **b** any hernia in physiological systems, cell types, and tissues, red columns have FDR < 5%.

associated loci, but most associated variants were observed for inguinal hernia and the two combined groups, both including inguinal hernia. We were able to perform a follow-up study for inguinal hernia with almost 24,000 additional cases from four study groups. In the resulting meta-analysis, 84 variants reached genome-wide significance, confirming all 53 variants initially at that level in the UK Biobank plus an additional 31 which included 22 new loci. Summing up all variants reaching genome-wide significance in any of the analyses, we identified 114 independently associated variants in 81 loci. Sex-specific analyses identified an additional 21 associated variants (14 for males/7 for females) which included 6 new loci. There are similarities between the different forms of hernia as all but diaphragmatic hernia is caused by weakness of the abdominal wall. The prevalence of diaphragmatic hernia varies between studies, as many cases do not experience symptoms and not get a diagnosis or any treatment[24]. Diaphragmatic hernia was also special as the majority of cases were assigned only from the questionnaire data,

leaving the possibility of reporting bias which could depend on the severity of associated health problems, especially gastro-esophageal reflux. The UK Biobank imaging data[25] will enable future studies on the prevalence of diaphragmatic hernia in a large subgroup of the cohort in relation to socio-demographic and health characteristics of the participants.

Several variants were identified in more than one GWAS scan and there were additional variants identified through the two combined groups. In total, 39 of the 86 variants reaching genome-wide significance in the UK Biobank data alone displayed the lowest P value in the analysis of any hernia (N = 11) or any hernia excluding diaphragmatic (N = 28), illustrating the power gained by this approach. With MultiPhen we assessed a set of associated hernia types for each variant and broke it down to the 81 loci. The majority of 46 loci were mainly associated with inguinal hernia, but there were 26 loci associated with multiple hernias (Fig. 3), strongly suggesting similarities in genetic mechanisms relevant for the different forms of hernia. On the

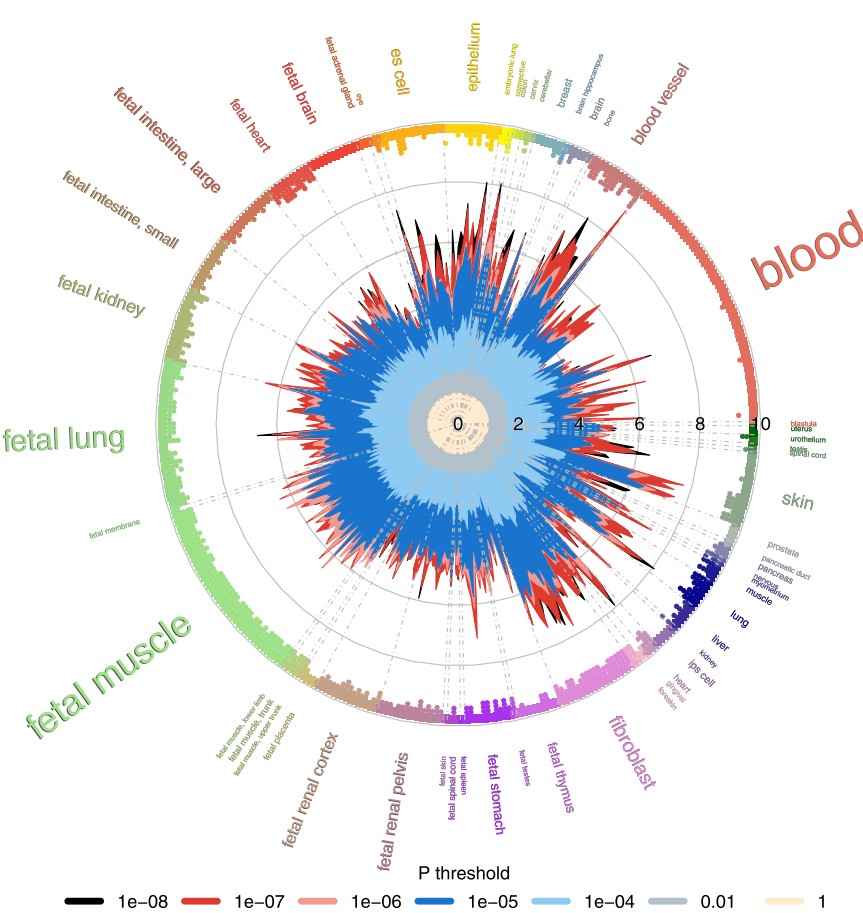

**Fig. 5 GARFIELD hotspots enrichment for inguinal hernia.** Enrichment wheel plot from the GARFIELD hotspots analysis for inguinal hernia. The radial spikes display the odds ratios for the tissues noted on the outside at seven $P$ value thresholds (color codes below the plot) for all ENCODE and Roadmap Epigenomics DHS cell lines. The dots just inside the outer circle denote significant (correcting for the number of effective annotations) GARFIELD enrichment separately for each of the five thresholds from $1 \times 10^{-4}$ to $1 \times 10^{-8}$.

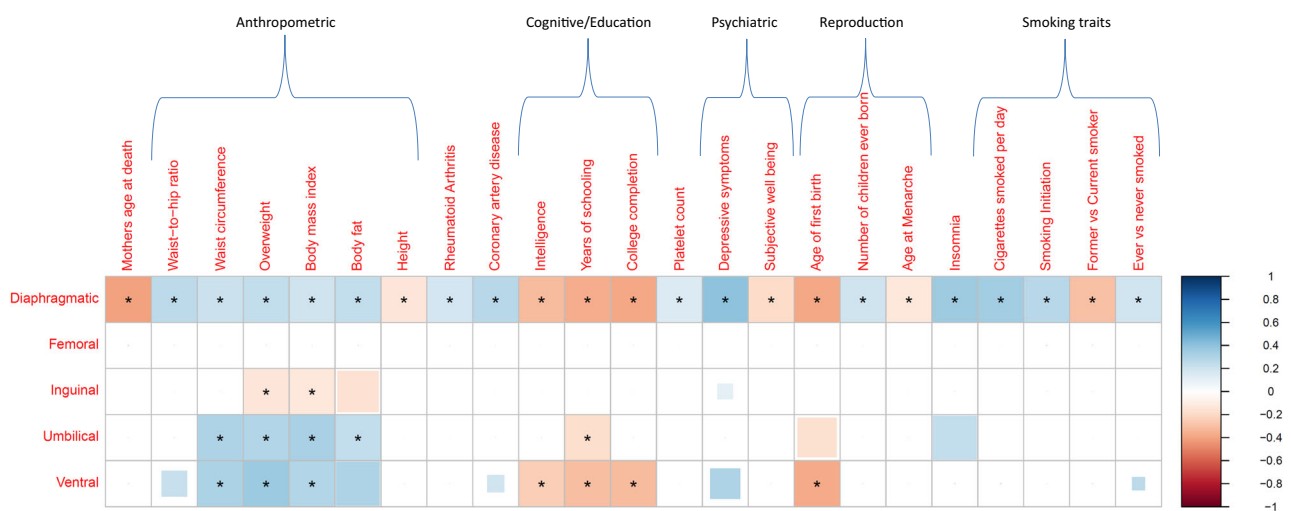

**Fig. 6 Genetic correlation analysis for the five forms of hernia.** Pairwise genetic correlations estimated via bivariate LD score regression for 245 traits. Only correlations that were significant after correction for multiple testing in at least one form of hernia are shown ($P < 2.04 \times 10^{-4}$, indicated by an asterisk). Positive genetic correlations are shown in blue, negative ones in red for all combinations where $P < 0.05$, with the size of the colored squares indicating the significance level (truncated at $P < 2.04 \times 10^{-4}$).

other hand, loci exclusively associated with one form of hernia could point to specific aspects of disease etiology. However, the huge differences in case numbers for the individual hernia types made it difficult to claim exclusive associations, as there was limited statistical power to detect attenuated signals related to small or moderate effects in the less frequent hernias. For the three less frequent hernias, there were only four genome-wide significant loci that appear to be exclusive (3 for umbilical, 1 for ventral, and 0 for femoral).

There were clear differences by sex for the association observed at some of the loci. However, factors like age at diagnosis or different sub-diagnoses might also play a role, which were not available or only available for a subset of the data.

Genetic correlation analysis between the hernia types substantiated the connections seen in our scans at the associated variants level and also confirmed the special role of diaphragmatic hernia as three of its four correlations were at the low end of the estimates. We identified further genetic links to other traits: positive associations between diaphragmatic hernia and several anthropometric traits are also seen in other hernias, for inguinal hernia the associations are reversed, i.e. an apparently protective effect of increased BMI, which is in line with previous epidemiological studies[3,26–29]. Negative correlations with educational attainment were observed for umbilical and ventral hernia. In the psychiatric field, positive associations between diaphragmatic/ventral hernia and depressive symptoms/major depression were found. Mendelian randomization analyses suggested a potential causal effect of depression on diaphragmatic hernia, but further studies are required to illuminate the relationship of these conditions.

Vulnerability of connective tissue is a plausible cause for herniation of the abdominal wall. The appearance of connective tissue depends on the physical and biochemical properties of the extracellular matrix (ECM)[30], and the ECM is composed of proteins including collagen and elastin. With this in mind, we reviewed the genes in the vicinity of the identified loci with FDR below 5% in the DEPICT gene prioritization and identified several good candidates: *ADAMTS6* was already discussed in the earlier inguinal hernia GWAS[7], due to the role of *ADAMTS* genes in encoding proteases which convert procollagen to collagen[31], *ADAMTS16* is another member of this gene family located in the multi-variant locus on chromosome 5p15.32 (tagging variant rs7715383), and *ADAMTSL3* on 15q25.2 (tagged by the intronic variant rs551555687) is from the *ADAMTS-like* family of genes and known to play a role in ECM processes[32]. *ELN* on 7q11.23 (tagging variant rs10224499) directly encodes elastin, the protein in the ECM relevant for the elasticity of many tissues and organs. *LOX* on 5q23.2 (tagged by the intronic variant rs3835066) encodes a lysyl oxidase protein, which is important for cross-linking collagen and elastin molecules[33]. Other genes with FDR < 5% in the DEPICT gene prioritization close to identified loci were from the transforming growth factor β family (*TGFB2*, *BMP5*, *BMP7*) encoding cell regulatory proteins with many basic functions, and *LTBP1* is relevant for TGF β activity and two more genes are coding ECM proteins (*FBLN2* and *MFAP4*). Further evidence for an involvement of several of these candidates was derived by an inguinal hernia-specific expression analysis resulting in nine genes meeting the significance threshold, eight of which were in loci reaching genome-wide significance (*ADAMTS6*, *DNAJC27*, *EFEMP1*, *FBLN2*, *HAND2*, *LOX*, *MFAP4*).

Overall, our study provides valuable insight into the genetics of herniation, identifying loci relevant for individual and multiple forms of hernia. Especially the fact that we identified additional loci by analyzing combined groups of hernia types can inspire future GWAS of other conditions that are also related amongst each other. The large number of associated loci for inguinal hernia point to several genes relevant for connective tissue homeostasis and general cell processes, which could be helpful to define genetic risk profiles or develop biomarkers indicating an increased vulnerability.

## Methods

**Study groups.** Additional details on the study groups are given in Supplementary Data 1.

**UK Biobank.** UK Biobank is a large-scale biomedical database and research resource containing genetic, lifestyle, and health information from half a million UK participants. Participants were not compensated for participation. Imputed genotypes and phenotype data are available for research[8]. In brief, only genetic markers present on both the UK BiLEVE and UK Biobank Axiom arrays, passing QC in all batches and with a missingness <5% and a minor allele frequency (MAF) > 0.0001 were considered, which resulted in a dataset with 670,739 autosomal markers in 487,442 samples[8]. We used the provided imputed genotype data and restricted the analyses to well-imputed variants (info > 0.8) with a MAF > 0.5% in 408,595 individuals of British ancestry. The questionnaire and hospital codes used for the five hernia types are listed in Supplementary Data 1. The combined groups of any hernia and any hernia excluding diaphragmatic hernia also included 4082 records of unspecific abdominal hernia repair from the interview on operations.

**FinnGen.** FinnGen (https://www.finngen.fi/en) is a large biobank study that aims to genotype 500,000 Finns and combine this data with longitudinal registry data including The National Hospital Discharge Registry, Causes of Death Registry and medication reimbursement registries, all of these linked by unique national personal identification codes. FinnGen includes prospective and retrospective epidemiological and disease-based cohorts as well as hospital biobank samples. Participants were not compensated for participation. The methods are described in detail online (https://finngen.gitbook.io/documentation/methods/cohort-description). In short, individuals were genotyped with Illumina and Affymetrix chip arrays (Illumina Inc., San Diego, and Thermo Fisher Scientific, Santa Clara, CA, USA). In sample-wise quality control, individuals with ambiguous gender, high genotype missingness (>5%), excess heterozygosity (+/− 4SD) and non-Finnish ancestry were excluded. In variant-wise quality control variants with high missingness (>2%), deviation from Hardy Weinberg equilibrium ($P < 1 \times 10^{-6}$) and minor allele count <3 were excluded. Genotype data were imputed using the population-specific SISu v3 imputation reference panel of 3775 whole genomes. The fifth data freeze (spring 2020) used in this study consists of 218,792 individuals. Inguinal hernia cases were identified via The National Hospital Discharge Registry and Causes of Death Registry. Association analysis of the imputed SNPs was performed with SAIGE[9], adjusting for age, sex, the first 10 principal components and genotyping batch. The analysis had 17,096 cases and 190,557 controls.

**Estonian Biobank.** Participants in the Estonian Biobank study were not compensated for participation. Samples were genotyped in Core Genotyping Lab of Institute of Genomics, University of Tartu using Illumina GSAv1.0, GSAv2.0, and GSAv2.0_EST arrays. Altogether 155,772 samples were genotyped and PLINK format files were created using Illumina GenomeStudio v2.0.4. Individuals were excluded from the analysis if their call rate was <95% or sex defined based on heterozygosity of X chromosome did not match sex in phenotype data. Variants were filtered by call-rate <95% and HWE $P$ value <1e$^{-4}$ (autosomal variants only). Variant positions were updated to build 37 and all variants were changed to be from TOP strand using tools and reference files provided in https://www.well.ox.ac.uk/~wrayner/strand/ webpage. Before imputation variants with MAF < 1% and indels were removed. QC was carried out using PLINK v1.9[34] and several in-house R and PERL scripts. After QC the dataset contained 154,201 samples for imputation. Prephasing was done using Eagle v2.3[35] and imputation was done using Beagle v.28Sep18.793[36] with effective population size ne = 20,000. Estonian population-specific imputation reference of 2297 whole-genome sequencing samples was used[37]. Analyses were restricted to individuals with European ancestry. Inguinal hernia cases were identified from electronic health records (ICD-10 codes K40, K40.0, K40.1, K40.2, K40.3, K40.4, and K40.9). Association analysis of the imputed SNPs was performed with SAIGE[9], adjusting for year of birth, sex, and the first six principal components. The analysis had 3151 cases and 138,028 controls.

**iPSYCH.** The iPSYCH study group included 78,050 individuals with GWAS data from the Illumina Infinium PsychChip v1.0[21]. Participants were not compensated for participation. Data cleaning involved filtering out SNPs that had a MAF of < 0.01 or deviated from Hardy-Weinberg equilibrium ($P < 1 \times 10^{-6}$). Structural variants, tri-allelic SNPs, non-autosomal SNPs, and SNPs that were strand ambiguous (A/T or C/G) or did not uniquely align to the genome were also excluded, resulting in an imputation backbone of 246,369 autosomal SNPs. A total of 364 samples failing basic QC (discordant sex information, more than 1% missing genotypes, abnormal heterozygosity, duplicate sample discordance) were also excluded. Based on the good quality SNPs, all individuals were phased in a single

batch using SHAPEIT3[38] and imputed in 10 batches using IMPUTE2[39] with reference haplotypes from the 1000 genomes project phase 3. Prior to analysis, we filtered out participants who were (1) of non-European ancestries, (2) related to another participant in the sample (corresponding to an IBD proportion >0.1875). We balanced the ratio of inguinal hernia cases to controls, so that it was the same within each of the 12 sex-specific groups of the iPSYCH study (5 psychiatric disorders: schizophrenia, affective disorder, ADHD, autism, and anorexia nervosa, as well as population controls). Inguinal hernia cases were identified from the National Patient Registry (ICD8 codes 550, 552, ICD-10 codes K40). Association analysis of imputed SNPs was performed with SAIGE[9], adjusting for year of birth, sex, and the first 6 PCs. The analysis had 2557 cases and 35,538 controls.

**Danish diverticular disease**. We used cases and controls from a Danish diverticular disease study, as described earlier[40]. Participants were not compensated for participation. Samples were genotyped with the Illumina OmniExpress-24v1-2_A1 array and we applied the following quality control steps, with PLINK (version 1.90b3o)[34], in order to sequentially remove (1) variant probes not uniquely aligned to the genome; (2) variants not present in dbSNP (version 146), (3) variants with conflicting reference alleles between Illumina and dbSNP; (4) variants with missingness >5%, (5) variants that failed Hardy-Weinberg test at $P < 1 \times 10^{-6}$, (6) variants with $p < 0.01$ in tests of differences in missingness between cases and controls; (7) variants with minor allele frequency <1%, (8) samples with discordant sex between registry records and genotypes; and (9) samples with missingness >2%. We also removed related individuals up until third degree relatives (kinship coefficient Phi > 0.05 estimated with KING[41] as implemented in VCFtools (version 0.1.12b)[42]). European ancestry outliers were removed based on principal component analysis of all our cases and controls combined with the European samples from the Human Genome Diversity Project[43] using the R package SNPRelate (version 1.2.0)[44]. Outliers were defined based on the first two principal components, excluding samples where the distance between at least one principal component and its mean was larger than 6 times the interquartile range. This outlier removal procedure was repeated until no more samples fulfilled the criterion for removal. After these steps, we imputed unobserved genotypes with phased haplotypes from the Haplotype Reference Consortium panel (version r1.1)[45] on the imputation server at the Wellcome Trust Sanger Institute (https://imputation.sanger.ac.uk/). Inguinal hernia cases were defined as having an ICD-10 code K40 (and sub-codes) and/or ICD-8 codes 550 and/or 552 (and-sub-codes). Association analysis of imputed SNPs was performed with SAIGE (version 0.29.4)[9], adjusting for year of birth, sex, and diverticular disease status. The analysis had 999 inguinal hernia cases and 7768 controls.

**Ethics statement**. We have complied with all relevant ethical regulations.

Participants of the UK Biobank have given written informed consent and the UK Biobank has approval from the North West Multi-centre Research Ethics Committee as a Research Tissue Bank (RTB), meaning that researchers do not require separate ethical clearance and can operate under the RTB approval (https://www.ukbiobank.ac.uk/learn-more-about-uk-biobank/about-us/ethics). This study was approved by the UK Biobank (project ID 33395).

Patients and control subjects in FinnGen provided informed consent for biobank research, based on the Finnish Biobank Act. Alternatively, separate research cohorts, collected prior the Finnish Biobank Act came into effect (in September 2013) and start of FinnGen (August 2017), were collected based on study-specific consents and later transferred to the Finnish biobanks after approval by Fimea, the National Supervisory Authority for Welfare and Health. Recruitment protocols followed the biobank protocols approved by Fimea. The Coordinating Ethics Committee of the Hospital District of Helsinki and Uusimaa (HUS) approved the FinnGen study protocol Nr HUS/990/2017. The FinnGen study is approved by Finnish Institute for Health and Welfare (permit numbers: THL/2031/6.02.00/2017, THL/1101/5.05.00/2017, THL/341/6.02.00/2018, THL/2222/6.02.00/2018, THL/283/6.02.00/2019, THL/1721/5.05.00/2019, THL/1524/5.05.00/2020, and THL/2364/14.02/2020), Digital and population data service agency (permit numbers: VRK43431/2017-3, VRK/6909/2018-3, VRK/4415/2019-3), the Social Insurance Institution (permit numbers: KELA 58/522/2017, KELA 131/522/2018, KELA 70/522/2019, KELA 98/522/2019, KELA 138/522/2019, KELA 2/522/2020, KELA 16/522/2020 and Statistics Finland (permit numbers: TK-53-1041-17 and TK-53-90-20). The Biobank Access Decisions for FinnGen samples and data utilized in FinnGen Data Freeze 6 include: THL Biobank BB2017_55, BB2017_111, BB2018_19, BB_2018_34, BB_2018_67, BB2018_71, BB2019_7, BB2019_8, BB2019_26, BB2020_1, Finnish Red Cross Blood Service Biobank 7.12.2017, Helsinki Biobank HUS/359/2017, Auria Biobank AB17-5154, Biobank Borealis of Northern Finland_2017_1013, Biobank of Eastern Finland 1186/2018, Finnish Clinical Biobank Tampere MH0004, Central Finland Biobank 1-2017, and Terveystalo Biobank STB 2018001.

The use of the Estonian Biobank data in this study was approved by the Research Ethics Committee of the University of Tartu (Approval number 288/M-18).

The Danish Scientific Ethics Committee, the Danish Data Protection Agency, and the Danish Neonatal Screening Biobank Steering Committee approved the iPSYCH study.

The Danish diverticular disease study was approved by the Scientific Ethics Committee of the Capital Region of Denmark (H-16016406) and the Danish Data Protection Agency. The Scientific Ethics Committee granted exemption from obtaining informed consent from participants as the study was based on the biobank material.

**Association analyses**. Initial genome-wide association studies for the 7 hernia groups were performed with SAIGE (version 0.29.4)[9], a generalized mixed model method allowing for related samples and thus maximizing power. Covariates were sex (only in the *all* analyses), year of birth, and the first six principal components calculated for the analyzed subset of the UK Biobank with British ancestry, $P$ values were adjusted by genomic control (Supplementary Data 1). To identify the most likely causal variants in the different GWAS scans, conditional and joined analyses[10] of the initial SAIGE results were performed with GCTA (version 1.26.0)[46] using the COJO option. We performed these analyses with thresholds of $5 \times 10^{-8}$ and $1 \times 10^{-5}$ to identify variants reaching genome-wide significance for the specific hernia group and to define a larger list of potentially associated variants for subsequent analyses, respectively. The meta-analysis for the inguinal hernia follow-up study and the heterogeneity analyses by sex for the five individual hernia types were performed with METAL (version 2011.03.25)[11]. Genome-wide significance was set per phenotype at $P < 5 \times 10^{-8}$, for inguinal hernia this was assessed from the combined analysis of the UK Biobank and the follow-up studies. We applied MultiPhen (version 2.0.2)[13], a method that models multiple phenotypes simultaneously by taking the genotype as dependent variable and performing ordinal regression, to a subset of 335,258 unrelated individuals (no third degree or closer relationship). The outcome is a linear combination of the five hernia types most strongly associated with the genotypes of a variant and we selected a $P$ threshold of 1% to define association for one of the five hernia groups. Potential new loci identified by MultiPhen were confirmed by linear regression models in R[14] with the allele count at the respective variant as outcome and the identified forms of hernia as explanatory variables (both $P < 5 \times 10^{-8}$ for the joint models).

**Pathway and enrichment analyses**. We applied several software packages and online tools to the association results to identify associated genes, genesets and pathways.

We performed MAGMA gene prioritization[15] as implemented in FUMA (version 1.3.6)[16] based on the complete GWAS scan results, and FUMA expression results for 54 GTEx tissues were derived from MAGMA analysis. All these analyses were run via the web on June 23 and 24, 2020, and the specifications and results are available in the public results section with IDs 424 to 430 (https://fuma.ctglab.nl/browse, author Frank Geller, titles UK Biobank followed by the respective hernia group).

We applied DEPICT (version 1 rel194)[17], which has three features: gene prioritization, pathway analysis and tissue/cell type enrichment analysis. Variants from the HLA region and chromosome X were not considered by the DEPICT analyses. To provide independent loci as input, we clumped variants with $P < 1 \times 10^{-5}$ in PLINK (version 1.90b3o, Supplementary Data 8).

Gene Network[18] (version 2.0, accession date 12/15/2020, https://www.genenetwork.nl/) uses 31,499 public RNA-seq sample to prioritize genes based on HPO terms. We generated GeneNetwork results for the HPO term inguinal hernia for the 192 genes studied in the DEPICT inguinal hernia gene prioritization.

GARFIELD (version 2)[19]—GWAS analysis of regulatory or functional information enrichment with LD correction, investigates regulatory and functional implications of GWAS findings. To focus the analyses on the variants identified in the conditional and joint GCTA-COJO analysis at $P < 1 \times 10^{-5}$, we removed all other variants with $P < 1 \times 10^{-5}$ from the input. The annotation step in GARFIELD still included all correlated variants at $r^2 > 0.8$ for the variants identified by GCTA-COJO.

**Genetic correlation, Mendelian randomization, heritability**. We investigated genetic correlation between the five hernia types as well as between them and 245 traits with a PubMed ID available in LD hub[47] by performing LD score regression.

Mendelian randomization analyses were carried out based on the 178 genetic variants independently associated ($P < 5 \times 10^{-8}$) with major depressive disorder (MDD) from the latest and largest GWAS of MDD[22] as instrumental variables, i.e. we used the effect estimates from the 23andMe European ancestry cohort (75,607 cases, 231,747 controls). To investigate causality of MDD with the different types of hernia, two-sample Mendelian randomization (MR) analysis was performed using the random-effects inverse-variance weighted method[48], implemented in the R package MendelianRandomization (version 0.5.0). Sensitivity analyses were performed with simple median (weighted median with simple median weighting[49], MR-Egger[50] (both also implemented in the MendelianRandomization package) and MR-PRESSO[51] (implemented in the MRPRESSO (version 1.0) R package) methods.

Heritability was estimated for the five hernia types by applying LD score regression[23,52], HESS (version 0.5.4-beta)[53] and LDAK (version 5.1)[54] to the UK Biobank discovery GWAS scans. Heritability was transformed to the liability scale with GEAR (https://sourceforge.net/p/gbchen/wiki/Hong%20Lee's%20Transformation%20for%20Heritability/).

**URLs**. UK Biobank website: www.ukbiobank.ac.uk
FinnGen website: https://www.finngen.fi/en
FinnGen methods: https://finngen.gitbook.io/documentation/methods/cohort-description.

Hernia phenotypes in mice from Mammalian Phenotype Ontology Annotations: www.informatics.jax.org/vocab/mp_ontology

Rayner strand alignment tool: www.well.ox.ac.uk/~wrayner/strand

Imputation server at the Wellcome Trust Sanger Institute: https://imputation.sanger.ac.uk

GEAR heritability: https://sourceforge.net/p/gbchen/wiki/Hong%20Lee's%20Transformation%20for%20Heritability/

**Reporting summary**. Further information on research design is available in the Nature Research Reporting Summary linked to this article.

## Data availability

The GWAS data from the UK Biobank analyses in this study have been deposited on the Danish National Biobank website (https://www.danishnationalbiobank.com/gwas). The remaining data generated in this study are provided in the Supplementary Information and Source Data file. Source data are provided with this paper.

## Code availability

Code for the analyses is available under HerniaCode at our GitHub site (https://github.com/FeenstraLab).

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

## Acknowledgements

We thank all participants and staff of the UK, FinnGen and Estonian biobanks and the iPSYCH and Danish diverticular disease study for their contribution to this research. This research has been conducted using data from UK Biobank, a major biomedical database (project ID 33395). The study was supported by grants from the Oak Foundation (OCAY-18-598), a Lundbeck Foundation Ascending Investigator grant (R313-2019-554) to B.F. The Danish National Biobank was established with the support of major grants from the Novo Nordisk Foundation, the Danish Medical Research Council and the Lundbeck Foundation. L.S. received support from a Carlsberg Foundation postdoctoral fellowship (CF15-0899). The FinnGen project is funded by two grants from Business Finland (HUS 4685/31/2016 and UH 4386/31/2016) and the following industry partners: AbbVie Inc., AstraZeneca UK Ltd, Biogen MA Inc., Celgene Corporation, Celgene International II Sàrl, Genentech Inc., Merck Sharp & Dohme Corp, Pfizer Inc., GlaxoSmithKline Intellectual Property Development Ltd., Sanofi US Services Inc., Maze Therapeutics Inc., Janssen Biotech Inc, and Novartis AG. Following biobanks are acknowledged for delivering biobank samples to FinnGen: Auria Biobank (www.auria.fi/biopankki), THL Biobank (www.thl.fi/biobank), Helsinki Biobank (www.helsinginbiopankki.fi), Biobank Borealis of Northern Finland (https://www.ppshp.fi/Tutkimus-ja-opetus/Biopankki/Pages/Biobank-Borealis-briefly-in-English.aspx), Finnish Clinical Biobank Tampere (www.tays.fi/en-US/Research_and_development/Finnish_Clinical_Biobank_Tampere), Biobank of Eastern Finland (www.ita-suomenbiopankki.fi/en), Central Finland Biobank (www.ksshp.fi/fi-FI/Potilaalle/Biopankki), Finnish Red Cross Blood Service Biobank (www.veripalvelu.fi/verenluovutus/biopankkitoiminta) and Terveystalo Biobank (www.terveystalo.com/fi/Yritystietoa/Terveystalo-Biopankki/Biopankki/). All Finnish Biobanks are members of BBMRI.fi infrastructure (www.bbmri.fi) and FinBB (https://finbb.fi/). The work of Estonian Genome Center, Univ. of Tartu has been supported by the European Regional Development Fund and grants No. GENTRANSMED (2014-2020.4.01.15-0012), MOBERA5 (Norface Network project no 462.16.107) and 2014-2020.4.01.16-0125. This study was also funded by the European Union through Horizon 2020 research and innovation program under grant no 810645 and through the European Regional Development Fund project no. MOBEC008 and Estonian Research Council Grant PUT1660. Data analyses with Estonian datasets were carried out in part in the High-Performance Computing Center of University of Tartu. The Initiative for Integrative Psychiatric Research (iPSYCH) was supported by grants from the Lundbeck Foundation (R165-2013-15320, R155-2014-1724, and R248-2017-2003), and the universities and university hospitals of Aarhus and Copenhagen, Denmark. The content is solely the responsibility of the authors and does not necessarily represent the official views of the funders.

## Author contributions

F.G. conceptualized the study and wrote the original manuscript draft. J.F. and F.G. coordinated the analyses. J.F., L.S., J.K., E.A., and F.G. analyzed the data. E.S. and H.U. provided data from the diverticular study. T.W. provided data from the iPSYCH study. T.E. and L.M. provided data from the Estonian Biobank study. A.P. and M.D. provided data from the FinnGen study. J.F., L.S., M.M., B.F., and F.G. interpreted the results. M.M., B.F., and F.G. supervised the study. All authors contributed to the revision of the manuscript and approved it for submission.

## Competing interests

J.F. currently works for Novo Nordisk A/S. The remaining authors declare no competing interests.

## Additional information

## iPSYCH Group

Thomas Werge[9,10,11,12], David M. Hougaard[9,17], Anders D. Børglum[9,18,19], Merete Nordentoft[9,10,20] & Preben B. Mortensen[9,21,22]

[17]Department for Congenital Disorders, Statens Serum Institut, Copenhagen DK-2300, Denmark. [18]Department of Biomedicine and iSEQ-Centre for Integrative Sequencing, Aarhus University, Aarhus, Denmark. [19]Center for Genomics and Personalized Medicine, Aarhus University, Aarhus, Denmark. [20]Copenhagen Research Centre for Mental Health, Mental Health Centre Copenhagen, Copenhagen University Hospital, Copenhagen, Denmark. [21]National Centre for Register-based Research, Aarhus University, DK-8210 Aarhus, Denmark. [22]Department of Biomedicine and iSEQ-Centre for Integrative Sequencing, Aarhus University, Aarhus, Denmark.

