## [Peer Review File · Nature Communications]

Title: Comprehensive genome-wide association study of different forms of hernia identifies more than 80 associated lociREVIEWER COMMENTS

Reviewer #1 (Remarks to the Author):

In this manuscript, the authors describe a series of genetic association analyses of abdominal hernia phenotypes, focusing on the genetic overlap of different anatomical subtypes. They utilize a suite of in silico tools to characterize the effects of the loci implicated in the association analyses, pointing to the important roles of collagen and elastin in hernia susceptibility. While some of the association results have been posted on the various UK Biobank pheweb sites, this study goes well beyond those findings with improved phenotype definitions and sex-specific analyses, something that is particularly key for inguinal and femoral hernia subtypes. As such, this is an important contribution to our understanding of genetic influences on hernia etiology.

The paper is well-written and clearly organized. There are many supplemental files that provide detailed results from the various analyses. If there is a moderate flaw here, it is that the authors spend a disproportionate amount of text listing the number of variants identified in various analyses compared to the time spent on explaining the novel biology discovered through this effort. This obscures some of the more interesting findings in the paper. The following suggestions are aimed at improving that aspect of the paper:

1. Hernias are associated with other human traits, and, of particular interest, this study finds a significant novel genetic correlation between major depressive disorder and diaphragmatic and ventral hernias. Genetic correlations don't address the direction of causation, whereas a Mendelian Randomization approach can. The authors should apply this approach, for example, using a polygenic risk score for major depression and testing its effect on the risk of the hernia subtypes studied here. Given the connection to the iPsych project, there may also be a possibility to test the converse (hernia predicting major depression).
2. The replication analysis is restricted to inguinal hernia, which is disappointing but also understandable as it is the most common form of hernia. While the lack of replication for other subtypes is a weakness, the overall set of analyses and clear genetic overlap between hernia subtypes tempers this weakness. That being said, one of the cohorts selected for replication could be problematic. A prior GWAS study of diverticular disease (PMID: 30177863) identified substantial overlap between hernias, in particular inguinal hernia. It is possible that the inclusion of this cohort could bias the replication results. In fact, the percentage of subjects that are cases is highest in the Danish diverticular disease cohort compared to the other cohorts (excluding the case/control balanced iPsych cohort). The authors should conduct a sensitivity analysis excluding this cohort to determine if it substantially changes the replication findings. A note describing their determination could be included in the discussion or in the supplemental text.
3. The opposite direction of effect for certain loci is quite interesting, as are the shared loci. Given the substantial overlap of loci across subtypes, it would be worthwhile to focus a short paragraph on the loci that act specifically on a single subtype, particularly the umbilical and ventral phenotypes, as this could point to some unique aspect of hernia etiology that differentiates these subtypes from the others.
4. A more minor concern is the age distribution for the replication cohorts. Subjects included are as

young as 2 years old. It is not clear whether congenital or childhood onset inguinal hernia is a comparable trait to adult-onset hernia. A comment in the discussion or supplemental text by the authors noting this potential weakness is warranted.

Reviewer #2 (Remarks to the Author):

Fadista et al. describe a massive dataset that sheds light into the genetic architecture of different types of hernias (diaphragmatic-, inguinal-, umbilical, ventral- and femoral hernias in decreasing order of sample size). In total, close to 65k cases and 350k controls were used in discovery, all from the UKB. Replication studies were performed in 4 additional studies with a total sample size of close to 25k cases and 370k controls.

The authors describe 81 independent loci and 26 were associated with >1 hernia type and pathway analyses show enrichment of collagen and elastin related signals.

The subject matter is interesting and the text is reasonably well written. The statistical methods are appropriate, in general (see below).

STRENGTHS

- Very large study on an unexplored subject where GWAS can help to disentangle pathways.
- World-experts in the field among the authors.
- Interesting findings with attractive illustrations by figures.

WEAKNESSES

- 1) Phenotype: There is likely to be a major reporting bias (e.g. what fraction of the diaphragmatic hernias are known?) that can be verified in the subgroup of participants of the UKB for who imaging data is available.
- 2) Adjustments: It is unclear what covariates were used in the replication cohorts. In UKB BMI does not appear to have been used, a potentially important point, notably because of the association of hernias with BMI and other anthropometrics.
- 3) Genotyping and quality control: there is very little information on this in the text, e.g. this reviewer could not find any information on allele frequency cutoffs for UKB for example, this information is given for the Estonian Biobank for example.
- 3) Quality control: QQ plots are not shown, genomic-control appears elevated for some traits (1.18).
- 4) Replication: It is unclear to this reviewer how many loci from the discovery analysis replicated in the replication studies for inguinal hernia. The authors state to have used "results from GCTA-COJO analyses of inguinal hernia at $P < 1 \times 10^{-5}$ as starting point", which is not comprehensible. This reviewer considers a replicated SNP if a significant p-values is reached in the replication study only, correcting for the number of SNPs submitted for replication.
- 5) Multiple testing: a large number of phenotypes and sex-specific analysis and groups were used. This needs to be reflected in the p-value cutoff used, particularly if all results are collapsed for further analyses.
- 6) Sex-specific analyses: it is unclear to this reviewer how the sex-specific analyses were done as the

discovery analysis was adjusted for sex. Was the sex-term significant for the SNPs tested ?

This reviewer has no critique regarding the MAGMA/FUMA - DEPICT - GN - GARFIELD and JAX analyses.

7) Heritability may be estimated by at least two methods.

MINOR

- The supplement includes >10 page description of the different contributors to FinnGen and their affiliations. This reviewer believes that such data belongs on a website, not into a scientific journal.

REVIEWER COMMENTS

Reviewer #1 (Remarks to the Author):

In this manuscript, the authors describe a series of genetic association analyses of abdominal hernia phenotypes, focusing on the genetic overlap of different anatomical subtypes. They utilize a suite of in silico tools to characterize the effects of the loci implicated in the association analyses, pointing to the important roles of collagen and elastin in hernia susceptibility. While some of the association results have been posted on the various UK Biobank pheweb sites, this study goes well beyond those findings with improved phenotype definitions and sex-specific analyses, something that is particularly key for inguinal and femoral hernia subtypes. As such, this is an important contribution to our understanding of genetic influences on hernia etiology.

The paper is well-written and clearly organized. There are many supplemental files that provide detailed results from the various analyses. If there is a moderate flaw here, it is that the authors spend a disproportionate amount of text listing the number of variants identified in various analyses compared to the time spent on explaining the novel biology discovered through this effort. This obscures some of the more interesting findings in the paper. The following suggestions are aimed at improving that aspect of the paper:

We thank the reviewer for the favorable comments. We think that there is value in stating some numbers to give the reader an overview of our findings, but the points raised by the reviewer cover interesting aspects of the study and we are happy to address them in the revised manuscript.

1. Hernias are associated with other human traits, and, of particular interest, this study finds a significant novel genetic correlation between major depressive disorder and diaphragmatic and ventral hernias. Genetic correlations don't address the direction of causation, whereas a Mendelian Randomization approach can. The authors should apply this approach, for example, using a polygenic risk score for major depression and testing its effect on the risk of the hernia subtypes studied here. Given the connection to the iPsych project, there may also be a possibility to test the converse (hernia predicting major depression).

Following the reviewer's suggestion, we have performed Mendelian randomization analyses of genetic variants associated with major depressive disorder (MDD) on the different forms of hernias and present the results in the *Results* section, Supplementary Table 7 and the Supplementary Note. We also planned to test the converse using the large GWAS on MDD from 23andMe that had provided the effect estimates for the MDD instrumental variables and applied for that dataset at the 23andMe Research Team last July. Following the initial evaluation of our request, we received a Data Transfer Agreement, which was signed and returned in the middle of October. After that, we got the message that we have to wait for the 23andMe signing official to fully execute the agreement. This has not happened yet despite several reminders. Also, the process will take additional time for the data transfer and approval of the manuscript by 23andMe. All the other parts of the revision have been done for months, so after consulting with the editor we decided to resubmit the manuscript in its current form. If the data from 23andMe should become available in the meantime, we will be happy to do this additional Mendelian randomization analysis and update the manuscript.

Anyway, the MR results indicated a potential causal effect of MDD on diaphragmatic hernia, but also saw signs of pleiotropy. Given that there are additional factors related to both, MDD and diaphragmatic

hernia (e.g. obesity), our conclusion is that MR is probably not perfectly suited to disentangle the interplay of these factors and longitudinal epidemiological studies are needed.

2. The replication analysis is restricted to inguinal hernia, which is disappointing but also understandable as it is the most common form of hernia. While the lack of replication for other subtypes is a weakness, the overall set of analyses and clear genetic overlap between hernia subtypes tempers this weakness. That being said, one of the cohorts selected for replication could be problematic. A prior GWAS study of diverticular disease (PMID: 30177863) identified substantial overlap between hernias, in particular inguinal hernia. It is possible that the inclusion of this cohort could bias the replication results. In fact, the percentage of subjects that are cases is highest in the Danish diverticular disease cohort compared to the other cohorts (excluding the case/control balanced iPsych cohort). The authors should conduct a sensitivity analysis excluding this cohort to determine if it substantially changes the replication findings. A note describing their determination could be included in the discussion or in the supplemental text.

We agree that replication studies for all forms of hernia would be desirable. However, there was a trade-off between including detailed analyses from other large cohorts and the number of analyzed phenotypes. Our approach was motivated both by the availability of large case numbers in the additional study groups and the fact that inguinal hernia showed the largest number of independent variants at $P < 1 \times 10^{-5}$, a reasonable cut-off for variants to follow up when sufficient sample sizes are at hand. We also hope to follow up the other forms of hernia in the future.

With regard to the inclusion of the diverticular disease (DD) study, we would like to mention that this study was not part of the initial GWAS, so that a bias towards DD loci when selecting the variants for follow-up can be ruled out. Furthermore, as in other studies that analyze phenotypes in cohorts recruited for other diseases, we adjusted for DD case-control status in the analyses.

The point of potential bias is nevertheless valid and we did a sensitivity analysis for clarification. The sensitivity analysis did not support a bias introduced by the DD cohort and we have added two sentences on DD in the *Results* section and provide the sensitivity analysis in the *Supplementary Note*. The detailed analysis is described in the Supplementary Note (section "Sensitivity analysis for the diverticular disease study") and Supplementary Table 3 (Sheet "sensitivity for DD").

3. The opposite direction of effect for certain loci is quite interesting, as are the shared loci. Given the substantial overlap of loci across subtypes, it would be worthwhile to focus a short paragraph on the loci that act specifically on a single subtype, particularly the umbilical and ventral phenotypes, as this could point to some unique aspect of hernia etiology that differentiates these subtypes from the others.

We agree that loci exclusively associated with one form of hernia could point to specific aspects of disease etiology. Here, it is important to keep in mind that the five forms of hernia were represented with case numbers ranging from 1,049 (femoral) to 31,143 (diaphragmatic), somehow reflecting the prevalence of these conditions in the UK Biobank. This implies huge differences in the statistical power of the five GWAS scans, meaning that variants reaching genome-wide significance in the less frequent hernias will easily be identified with even attenuated effects in the frequent groups of diaphragmatic and inguinal hernia, whereas there is limited power to detect small and moderate effects in the less frequent hernias. Taking a closer look at the less frequent hernias, they all include the two loci on chromosome 1 (also observed for inguinal) and chromosome 2 (also observed in inguinal and diaphragmatic). Overall, there are only 3 of 9 genome-wide significant umbilical hernia loci that appear to be exclusive, 1 of 3 for

ventral and 0 of 2 for femoral. Looking at the three loci for umbilical hernia, there were no genes belonging to the same pathway, so the limited number of associated loci limits the potential of detecting new mechanisms. Therefore, we have added the following sentences to the Discussion:

“On the other hand, loci exclusively associated with one form of hernia could point to specific aspects of disease etiology. However, the huge differences in case numbers for the hernia types made it difficult to claim exclusive associations, as there was limited statistical power to detect attenuated signals related to small or moderate effects in the less frequent hernias. For the three less frequent hernias, there were only four genome-wide significant loci that appeared to be exclusive in the MultiPhen analysis (3 for umbilical, 1 for ventral and 0 for femoral).”

However, we think that the pathway analyses for the combined group and inguinal hernia provide valuable insight that probably also has implications for the less frequent hernia groups.

4. A more minor concern is the age distribution for the replication cohorts. Subjects included are as young as 2 years old. It is not clear whether congenital or childhood onset inguinal hernia is a comparable trait to adult-onset hernia. A comment in the discussion or supplemental text by the authors noting this potential weakness is warranted.

We now include three sentences in the discussion to clarify that early onset cases need further study: “Our study did not consider age at diagnosis in the case definition (which was not available for the UK Biobank diagnoses and questionnaires), so that some cases in the initial and replication groups will be congenital / childhood onset. However, except for the iPSYCH study, the majority of the participants in the studies had an age well above 40 which meant that the available electronic health records did not cover their childhood. Specific studies with substantial numbers of early onset cases are needed to clarify whether the identified variants have an effect in this age spectrum.”

Reviewer #2 (Remarks to the Author):

Fadista et al. describe a massive dataset that sheds light into the genetic architecture of different types of hernias (diaphragmatic-, inguinal-, umbilical, ventral- and femoral hernias in decreasing order of sample size). In total, close to 65k cases and 350k controls were used in discovery, all from the UKB. Replication studies were performed in 4 additional studies with a total sample size of close to 25k cases and 370k controls.

The authors describe 81 independent loci and 26 were associated with >1 hernia type and pathway analyses show enrichment of collagen and elastin related signals.

The subject matter is interesting and the text is reasonably well written. The statistical methods are appropriate, in general (see below).

STRENGTHS

- Very large study on an unexplored subject where GWAS can help to disentangle pathways.
- World-experts in the field among the authors.
- Interesting findings with attractive illustrations by figures.

We thank the reviewer for the positive feedback and for spotting some weaknesses. We have addressed these points to improve the manuscript.

WEAKNESSES

@1) Phenotype: There is likely to be a major reporting bias (e.g. what fraction of the diaphragmatic hernias are known?) that can be verified in the subgroup of participants of the UKB for who imaging data is available.

We were in correspondence with Dr. Littlejohns (first and corresponding author on the UK Biobank Imaging Study paper¹) and Dr. Lacey from the UK Biobank scientific team. They both agreed that it should be possible to diagnose diaphragmatic hernia from the abdominal MRIs, but they are not aware of any studies on this. The imaging data will be made available in the near future via the UK Biobank cloud system, but we have to admit that diagnostics based on the imaging data is beyond the scope of our study. We have added three sentences on diaphragmatic hernia to the discussion: *“The prevalence of diaphragmatic hernia varies between studies, as many cases do not experience symptoms and not get a diagnosis or any treatment². Diaphragmatic hernia was also special as the majority of cases was assigned only from the questionnaire data, leaving the possibility of reporting bias which could depend on the severity of associated health problems, especially gastroesophageal reflux. The UK Biobank imaging data¹ will enable future studies on the prevalence of diaphragmatic hernia in a large subgroup of the cohort in relation to socio-demographic and health characteristics of the participants.”*

@2) Adjustments: It is unclear what covariates were used in the replication cohorts. In UKB BMI does not appear to have been used, a potentially important point, notably because of the association of hernias with BMI and other anthropometrics.

In line with the initial GWAS on inguinal hernia³, we did not adjust for BMI in our analyses. Our approach has the advantage that the involvement of BMI in the etiology of hernias is reflected in the genetic correlation between our GWAS (not adjusted for BMI) and the BMI / obesity GWAS. Furthermore, Aschard et al. generally advised against adjusting for heritable covariates as they can bias effect estimates: *“Overall, when the goal is to identify genetic variants that are directly associated with a primary outcome, we were unable to identify an alternative approach that adjusts for a covariate and leads to unbiased effect estimates for a heritable covariate that is associated with the tested variant (see Appendix D). Therefore, unless we know with certainty that the tested variant does not influence the covariate, we recommend that the inclusion of such heritable covariates in the model should be avoided. Given evidence for a large number of pleiotropic genes across complex traits, it seems unlikely that any heritable covariates with a complex genetic architecture, e.g., BMI or WHR, will fulfill that condition. Including such covariates in the absence of a strong prior knowledge on the pathophysiology is therefore likely to lead to biased effect estimates”⁴.*

The replication cohorts were analyzed similarly to the initial UK Biobank data with minor study specific modifications. The details are given in the revised *Online Methods* section.

3) Genotyping and quality control: there is very little information on this in the text, e.g. this reviewer could not find any information on allele frequency cutoffs for UKB for example, this information is given for the Estonian Biobank for example.

Also with regard to the previous point, we have extended the paragraphs on the study groups in the *Online Methods* section to provide additional information on the analyses and QC performed in the individual groups. Additionally, for all study groups references with more comprehensive study information are provided.

3) Quality control: QQ plots are not shown, genomic-control appears elevated for some traits (1.18).

We now provide QQ plots in the Supplementary Note. We would like to mention that all our presented results and subsequent analyses are based on P values adjusted for the respective genomic control factor. Furthermore, it is common that many loci underlie genetic variation for a condition with polygenic inheritance, implying that a substantial proportion of the genome should show inflation of the test statistic at large sample size and the observed inflation is in line with theoretical considerations and empirical observations⁵.

4) Replication: It is unclear to this reviewer how many loci from the discovery analysis replicated in the replication studies for inguinal hernia. The authors state to have used "results from GCTA-COJO analyses of inguinal hernia at $P < 1 \times 10^{-5}$ as starting point", which is not comprehensible. This reviewer considers a replicated SNP if a significant p -value is reached in the replication study only, correcting for the number of SNPs submitted for replication.

We apologize that the term "replication" caused confusion in the manuscript. It is commonly used for the second stage in two-stage approaches in GWAS, even if the goal is not to assess significance only based on the replication study in the way the reviewer describes. However, it is an efficient strategy to take independent genetic variants associated at a certain threshold P to a second stage with additional study groups. Then, the best strategy is to assess significance based on the combined analysis of all data⁶. For clarification, we rephrased the analyses of inguinal hernia in additional study groups as follow-up study and explain that we use the threshold of 5×10^{-8} in the combined analysis to determine significantly associated variants. We further describe the good agreement between the initial results and the results in the follow-up studies in the *Results* section.

5) Multiple testing: a large number of phenotypes and sex-specific analysis and groups were used. This needs to be reflected in the p -value cutoff used, particularly if all results are collapsed for further analyses.

The reviewer is right that some studies strictly adjust for multiple testing by dividing the threshold by the number of analyzed phenotypes. Our study considers five related hernias and two combinations of these cases. Therefore, we apply a phenotype-wise genome-wide significance threshold of 5×10^{-8} , a strategy also used in e.g. a GWAS of five lipid traits⁷ or four smoking-related traits⁸. Our experience with the inguinal hernia follow-up study was that all 53 variants with a $P < 5 \times 10^{-8}$ in the UK Biobank analysis also were below the threshold in the final combined analysis (51 were even below 7.14×10^{-9} , a strict Bonferroni threshold accounting for 7 analyses). Generally, the 5×10^{-8} in GWAS is considered rather conservative, especially in studies of polygenic traits with large sample size⁵, and applying a stricter threshold would reduce the statistical power of the analyses. Based on results from the GWAS catalog even a relaxation of the threshold was discussed⁹. We saw similar results for inguinal hernia, where 5 of 6 variants with P in the interval $(5 \times 10^{-8}, 1 \times 10^{-7})$ in the UKB data had $P < 5 \times 10^{-8}$ in the final combined analysis (4 were even below 7.14×10^{-9}). To illustrate that the vast majority of our findings is not derived

by a too liberal threshold, we now state the number of variants reaching the conservative threshold adjusting for the 7 analyses in the *Results*, but continue with the variants identified at 5×10^{-8} , similar to a study of 143 dietary habits¹⁰.

Sex-specific analyses are usually considered as a means to delineate the findings for the whole group, thus further adjustment for these analyses was not made. In our study it is noteworthy that a few variants only reached genome-wide significance in one of the sex-specific analyses and not the combined analyses of both sexes. However, we collapsed the loci and did subsequent gene and pathway analyses only for the variants identified in the combined analyses of females and males.

6) Sex-specific analyses: it is unclear to this reviewer how the sex-specific analyses were done as the discovery analysis was adjusted for sex. Was the sex-term significant for the SNPs tested ?

Sex-specific analyses were done by only analyzing individuals of the respective sex. This was stated in the "Genome-wide association scans in the UK Biobank" section and we rephrased the sentence in the *Association analyses* section for further clarification: Covariates were sex (only in the all analyses), year of birth and the first six principal components calculated for the analyzed subset of the UK Biobank with British ancestry, ...

This reviewer has no critique regarding the MAGMA/FUMA - DEPICT - GN - GARFIELD and JAX analyses. Thank you.

7) Heritability may be estimated by at least two methods.

We now also include results based on two additional methods, HESS¹¹ and LDKA¹². The results now read: *"We estimated heritability for the five forms of hernia on the liability scale with three different methods (Supplementary Table 7). The estimates for LD score regression²² were 12.3% (95% CI: 9.8% - 14.7%) for inguinal hernia and ranging from 9.0% (95% CI: 5.8% - 12.3%) for ventral hernia to 12.8% (95% CI: 9.5% - 16.1%) for umbilical hernia, the results for the other two methods were lower in most instances."*

MINOR

- The supplement includes >10 page description of the different contributors to FinnGen and their affiliations. This reviewer believes that such data belongs on a website, not into a scientific journal.

Nature communications requests that the contributors from groups in the author list are given in the *Supplementary Note* (which is only available online), and this is probably a good idea as some external web link could easily vanish over time. However, we can also see that the list is very long for our article, so that readers only interested in the scientific part of the *Supplementary Note* could end up printing out many pages with the information on contributors. We now provide a *Supplementary Note Contributors* with the respective information and hope that this solution is fine with both of you, the reviewer and Nature Communications.

1. Littlejohns, T. J. *et al.* The UK Biobank imaging enhancement of 100,000 participants: rationale, data collection, management and future directions. *Nature Communications* (2020) doi:10.1038/s41467-020-15948-9.

2. Roman, S. & Kahrilas, P. J. The diagnosis and management of hiatus hernia. *BMJ (Online)* (2014) doi:10.1136/bmj.g6154.
3. Jorgenson, E. *et al.* A genome-wide association study identifies four novel susceptibility loci underlying inguinal hernia. *Nat. Commun.* **6**, 10130 (2015).
4. Aschard, H., Vilhjálmsson, B. J., Joshi, A. D., Price, A. L. & Kraft, P. Adjusting for heritable covariates can bias effect estimates in genome-wide association studies. *Am. J. Hum. Genet.* (2015) doi:10.1016/j.ajhg.2014.12.021.
5. Yang, J. *et al.* Genomic inflation factors under polygenic inheritance. *Eur. J. Hum. Genet.* (2011) doi:10.1038/ejhg.2011.39.
6. Skol, A. D., Scott, L. J., Abecasis, G. R. & Boehnke, M. Joint analysis is more efficient than replication-based analysis for two-stage genome-wide association studies. *Nat. Genet.* (2006) doi:10.1038/ng1706.
7. Willer, C. J. *et al.* Discovery and refinement of loci associated with lipid levels. *Nat. Genet.* (2013) doi:10.1038/ng.2797.
8. Erzurumluoglu, A. M. *et al.* Meta-analysis of up to 622,409 individuals identifies 40 novel smoking behaviour associated genetic loci. *Mol. Psychiatry* (2020) doi:10.1038/s41380-018-0313-0.
9. Panagiotou, O. A. *et al.* What should the genome-wide significance threshold be? Empirical replication of borderline genetic associations. *Int. J. Epidemiol.* (2012) doi:10.1093/ije/dyr178.
10. Cole, J. B., Florez, J. C. & Hirschhorn, J. N. Comprehensive genomic analysis of dietary habits in UK Biobank identifies hundreds of genetic associations. *Nat. Commun.* (2020) doi:10.1038/s41467-020-15193-0.
11. Shi, H., Kichaev, G. & Pasaniuc, B. Contrasting the Genetic Architecture of 30 Complex Traits from Summary Association Data. *Am. J. Hum. Genet.* **99**, 139–153 (2016).
12. Speed, D., Holmes, J. & Balding, D. J. Evaluating and improving heritability models using summary statistics. *Nat. Genet.* **2020 524 52**, 458–462 (2020).

REVIEWER COMMENTS

Reviewer #2 (Remarks to the Author):

Fadista et al. have made great efforts to revise the manuscript and the text has further increased in clarity. This is a major piece of work for the field and the authors are to be congratulated.

This reviewer does not agree with all the statements (e.g. that adjusting for co-variables in GWAS is not recommended), but I believe that the analyses are nevertheless valid (sensitivity analyses could have been conducted).

The remaining points being of distress to this reviewer are that

1) Only a fraction of the discovered loci were truly replicated (multiple-test adjusted significant result in phase 1 and then multiple-test adjusted significant result in phase 2). The authors correctly do not use the word "replication" in the text. But I believe that for transparency, the truly replicated loci could be marked in one of the results tables and a statement could be included describing that "conservative" replication did yield x-number of loci only.

2) The QQ-plots look unusual and show early major inflation. Why are observed p-values capped at 10^{-10} ?

Congratulations to this piece of work !

REVIEWER COMMENTS

Reviewer #2 (Remarks to the Author):

Fadista et al. have made great efforts to revise the manuscript and the text has further increased in clarity. This is a major piece of work for the field and the authors are to be congratulated. This reviewer does not agree with all the statements (e.g. that adjusting for co-variables in GWAS is not recommended), but I believe that the analyses are nevertheless valid (sensitivity analyses could have been conducted).

Setting up a comprehensive GWAS project includes many choices and we think that no two groups would do it exactly the same way. Thus, it is natural that the reviewer would have done some things differently, but we are happy that the reviewer came to the conclusion that our analyses are nevertheless valid.

The remaining points being of distress to this reviewer are that

1) Only a fraction of the discovered loci were truly replicated (multiple-test adjusted significant result in phase 1 and then multiple-test adjusted significant result in phase 2). The authors correctly do not use the word "replication" in the text. But I believe that for transparency, the truly replicated loci could be marked in one of the results tables and a statement could be included describing that "conservative" replication did yield x-number of loci only.

We have added the column replication to Supplementary Table 3, Sheet meta ING all, which has the three outcomes "yes", "no" and "NA", describing for the 41 SNPs with $P < 7.14 \times 10^{-9}$ (adjusting for 7 analyzed phenotypes) whether the combined analysis of the follow-up studies yielded a $P < 1.22 \times 10^{-3} = 0.05/41$ ("yes") or not ("no"). Eight of the 41 SNPs did not fulfill the criterion for conservative replication, but for all eight the effect of the combined follow-up studies went the same direction as in the UK Biobank and six of them reached nominal significance ($P < 0.05$). For SNPs with the initial $P > 7.14 \times 10^{-9}$ replication was not applicable ("NA"). We have added the following sentence in the Results section: *"Applying a conservative replication approach demanding a $P < 7.14 \times 10^{-9}$ (adjusting for 7 analyzed phenotypes without taking the correlation between them into account, reached by 41 variants) in the UK Biobank GWAS scan, and a $P < 1.22 \times 10^{-3}$ (0.05 divided by the 41 variants taken forward) in the combined follow-up studies, would result in 33 variants fulfilling both of these criteria."*

2) The QQ-plots look unusual and show early major inflation. Why are observed p-values capped at 10^{-10} ?

As mentioned in our initial reply, it is common that many loci underlie genetic variation for a condition with polygenic inheritance, and the observed inflation is in line with theoretical considerations and empirical observations of inflated test statistics at large sample sizes¹.

Here, it has to be kept in mind that the associated variants are not represented by single points in the QQ-plots, as all variants in linkage disequilibrium with these variants also have increased test statistics. Taking inguinal hernia as an example, it can be seen from the QQ-plot that there are thousands of variants with $P < 5 \times 10^{-8}$, reaching all the way to 10^{-3} for the expected values on the x-axis. All these variants are in the genomic regions surrounding the 53 independently associated variants with $P < 5 \times 10^{-8}$ in Supplementary Table 3 (sheet META INGall). To illustrate the effect of associated loci increasing the number of variants at less stringent P levels, we took a closer look at the 4,604 SNPs in the region (+/-

500 kb) of the variant with the strongest signal (lowest observed adjusted $P = 2.91 \times 10^{-68}$ for rs59985551). The counts (and percentages of the 4,604 SNPs in the region) per threshold are

$P_{adjusted} < 5 \times 10^{-8}$: 527 (11.4%)

$P_{adjusted} < 1 \times 10^{-4}$: 820 (17.8%)

$P_{adjusted} < 0.01$: 1,403 (30.5%)

Furthermore, also at the empirical level, a recent GWAS on lipid levels² presented genomic control stratified by minor allele frequency (Supplementary Table 2 in that paper), where they observed λ values between 1.45 and 2.08 for the common variants ($\geq 5\%$ minor allele frequency) in the large European dataset. We added the respective numbers to Supplementary Table 1 (sheet Genomic control) and see a similar pattern with the highest λ values observed in the common variants, but only reaching from 1.01 to 1.24. For our study, the majority of variants (64%) were from the common category, and thus the overall λ is also relatively close to the λ for this subset, whereas the lipid study apparently included more rare variants and therefore had a substantially lower fraction of common variants. We have rephrased the referring sentence in the Results section: "*P values were adjusted by genomic control, ranging from 1.00 (femoral) to 1.18 (any hernia), and the increased genomic control factors were largely driven by common variants (minor allele frequency $\geq 5\%$, Supplementary Table 1), which also accounted for the vast majority of variants with $P < 5 \times 10^{-8}$.*"

Finally, we would like to mention that the highest λ at 1.18 was observed for the combined group any hernia, which apart from having the largest sample size also combines the five different forms of hernia and therefore not only has signals for any hernia but also attenuated signals of variants only associated with one of the individual forms of hernia, especially inguinal and diaphragmatic (which can also be seen in the Manhattan plots, Figure 2).

The reasoning behind the truncated observed P values in the QQ-plots is given in the description of the Figures on page 1 of the Supplementary Note: "*Figures display expected vs. observed initial (red) and adjusted (blue) $-\log_{10}(P)$. Observed P are truncated at $P = 1 \times 10^{-10}$ for higher resolution of the area where the observed values start to exceed the expected ones.*" In our view, the truncated QQ-plots make it also easier to compare the plots over the different GWAS, but to provide the full picture we have now added not truncated versions of the QQ-plots below the truncated ones.

Congratulations to this piece of work !

Thank you!

References

1. Yang, J. *et al.* Genomic inflation factors under polygenic inheritance. *Eur. J. Hum. Genet.* (2011) doi:10.1038/ejhg.2011.39.
2. Graham, S. E. *et al.* The power of genetic diversity in genome-wide association studies of lipids. *Nat.* 2021 6007890 600, 675–679 (2021).